# Direct medical cost of adult Covid-19 ınpatients and ıts determinants at a university hospital

**Medine Gözde Üstündağ[1], Nazım Ercüment Beyhun[2], Murat Topbaş[2], Sevil Turhan[2]**

**1** Kars Provincial Health Directorate, Public Health Services, Kars, Türkiye, **2** Department of Public Health, Karadeniz Technical University Faculty of Medicine, Trabzon, Türkiye

**Citation:** Üstündağ MG, Beyhun NE, Topbaş M, Turhan S (2025) Direct medical cost of adult Covid-19 ınpatients and ıts determinants at a university hospital. PLoS ONE 20(4): e0319762. https://doi.org/10.1371/journal.pone.0319762

## Abstract

### Background and objectives

The aim of this study is to determine the direct medical costs of adult cases diagnosed with Covid-19 and hospitalized at Karadeniz Technical University Faculty of Medicine Farabi Hospital (KTUFMFH).

### Methods

This is a cost of illness study. The direct medical costs of adult cases who were hospitalized and treated for Covid-19 at KTUFMFH between 13.03.2020-12.03.2021 were examined. The study was retrospectively, on hospital perspective, using a prevalence-based approach and a bottom-up costing method. Per case costs, distribution of costs, and factors were examined for 113 cases. The determinants of the costs were evaluated via logistic regression analysis.

### Results

The median cost per case was 263.55 (min: 28.30 – max: 18,947.60) USD [229.55 (min: 23.90 – max: 15,633.67) EUR]. One-day increase in hospitalization (O.R. = 2.600 and %95 C.I. = 1.576-4.289), the presence of chronic disease (O.R. = 15.130 and %95 C.I. = 1.644-139.216) and receiving oxygen inhalation therapy (O.R. = 15.238 and %95 C.I. = 1.784-130.157) have been identified to increase medical cost of Covid-19 inpatients.

### Conclusion

Covid-19 hospital costs are of significant economic burden. The actions in order to lower the severity of the disease may play role to lower the medical costs in future coronavirus related pandemics.

## 1. Introduction

Coronavirus disease 2019 (Covid-19) is a viral respiratory illness caused by a novel coronavirus [1–3]. The clinical course of the disease ranges from asymptomatic to acute respiratory

**Data availability statement:** All relevant data are within the manuscript and its Supporting information files.

**Funding:** The author(s) received no specific funding for this work.

**Competing interests:** The authors have declared that no competing interests exist.

failure and even death. While 15% of symptomatic cases have a severe course, critically serious disease develops in 5% [4,5]. There is no specific treatment for the disease accepted all over the world. According to the clinical characteristics of the cases, drugs, fluids, blood and nutritional products are used in the treatment, and different therapeutic procedures such as interventional and medical applications are performed [2,3].

In the fight against Covid-19, disease prevention methods, drug and vaccine development studies, diagnosis and treatment create a great burden on the health systems of countries. The fact that the disease can progress in different severity among individuals causes variation in medical costs. Factors such as the lack of specific treatment for the disease and the use of different drugs together, the accompanying comorbid conditions or secondary infections, the need for hospitalization, and the need to use medical devices in severe cases increase the direct medical cost of the disease [6].

Covid-19 causes serious workforce loss by causing many people to become ill and die in a short time. In addition, it is an important public health problem due to the high cost burden it brings to countries [6]. In the study in Brazil, the total hospital cost of 3,254 hospitalizations was 41,122,173 USD [7]. A study conducted in China estimated the total hospital cost to be 494,202,120 USD based on the total number of Covid-19 cases in China (81,416) as of 21.03.2020 [8]. A simulation model developed by Bartsch and colleagues in the United States estimated that a single symptomatic SARS-CoV-2 infection had a median direct medical cost of 3,045 USD when only costs incurred during infection were included [9]. In this context, determining the medical costs of Covid-19 and distributing the cost to subgroups will be an important guide in the management of the use of limited medical resources.

The aim of this study is to determine the direct medical costs of adult patients diagnosed with Covid-19 and hospitalized in Karadeniz Technical University Faculty of Medicine Farabi Hospital (KTUFMFH) in the first year of the pandemic, their distribution to cost subgroups, and the factors affecting the costs.

## 2. Materials and methods

### 2.1. Research method in terms of disease cost analysis

The research is a descriptive type of disease cost analysis (cost of illness analysis). It was conducted retrospectively from a hospital perspective, using a prevalence-based approach, and a bottom-up costing method. The research is in the disease-specific study group.

There are two approaches to estimate the cost of illness: the bottom-up approach and the top-down approach. The bottom-up approach multiplies the average cost of the illness per patient by the prevalence of the illness. The top-down approach uses aggregated data and a population-attributable fraction to assign a percentage of total expenditure to the disease of interest [10]. In order for the study to reflect the costs more realistically, real costs were taken as basis and calculations were made using the bottom-up method based on individual unit prices.

Direct Costs are the expenses made for the resources used in the provision of health services. Healthcare workers' salaries, consumables used, drug expenses, machinery and equipment usage, operating expenses, physical infrastructure expenses; hospital hospitality services, laboratory services, other capital costs, maintenance and repair of buildings, vehicles and tools are examples of these costs. Direct costs are divided into two groups as direct medical and direct non-medical costs. Direct medical costs are the costs that include the purchasable services resulting from hospital treatment. Costs such as hospitalization costs, outpatient services, emergency room care costs, home care costs, rehabilitation service costs, physician and other health care workers' salaries, diagnostic tests, drug and

medical supplies costs are included in this group. Direct Non-Medical Costs are the costs of the resources required to benefit from health services provided in the health sector. Indirect Costs are the economic losses caused by the event in question due to the disease in areas other than the health sector, i.e. the cost of the change in the person's productivity. Unmeasurable Costs are the psychological costs reflected from the health sector to patients and their families [11,12].

Direct medical costs (physician examination fees, laboratory tests, radiological imaging methods, therapeutic procedures, bed expenses, drugs, non-drug therapeutic products and consumables) were evaluated in the study.

In this study, costs related to COVID-19 diagnosis and treatment were calculated based on the unit cost of each procedure, including the fees included in the package payment form, and on the actual costs.

## 2.2. Place, population and sample

The research was conducted at KTUFMFH. The hospital is a university hospital serving in Trabzon with over 800 beds and over 2,500 staff [13]. The universe is adult patients who were hospitalized with the diagnosis of Covid-19 in KTUFMFH between 13.03.2020 and 12.03.2021. We planned the study for one year to equalize seasonal differences and ensure diversity of applications. The costs incurred until the end of the hospitalization period of the patients were evaluated. The sample was not selected, and all patients who met the inclusion criteria were included in the study.

## 2.3. Inclusion and exclusion criteria

With the emergence of Covid-19, a Covid-19 clinic and Covid-19 service were established in our hospital. All patients suspected of Covid-19 received outpatient treatment from the emergency or Covid-19 clinic or infectious diseases clinic. All hospitalizations due to Covid-19 were made to the Covid-19 services or infectious diseases ward or intensive care unit.

### 2.3.1. Inclusion criteria.

- Diagnosed with Covid-19 in one of the emergency or Covid-19 or infectious diseases outpatient clinics in KTUFMFH (In order to include all costs of the patient from the beginning to the end of the process and because these costs of those diagnosed outside the hospital were not accessible, only patients diagnosed at KTUFMFH were included in the study.),

- Inpatient treatment started between 13.03.2020 and 12.03.2021 and lasted at least 24 hours,

- Hospitalization in at least one of the emergency, Covid-19, infectious diseases services or intensive care units (ICU) and no hospitalizations other than these services,

- Patients over the age of 18 who were registered with the ICD-10 diagnostic code (U07.3) specified in the Hospital Information Management System (HIMS) were included in the study.

### 2.3.2. Exclusion criteria.
Patients who were not registered to the services specified in KTUFMFH, whose hospitalization started after 12.03.2021, who were diagnosed with Covid-19 outside of KTUFMFH or who received inpatient treatment were not included in the study.

Patients were identified from the Hospital Information Processing Center (HIPC). Considering the longest resultant time of SARS-CoV-2 diagnostic tests, the outpatient clinic

applications of these patients within the last 72 hours were examined. In case of applications other than the specified outpatient clinics, only the cost of the specified outpatient clinic applications was taken into consideration. The study was completed with 113 patients (Fig 1).

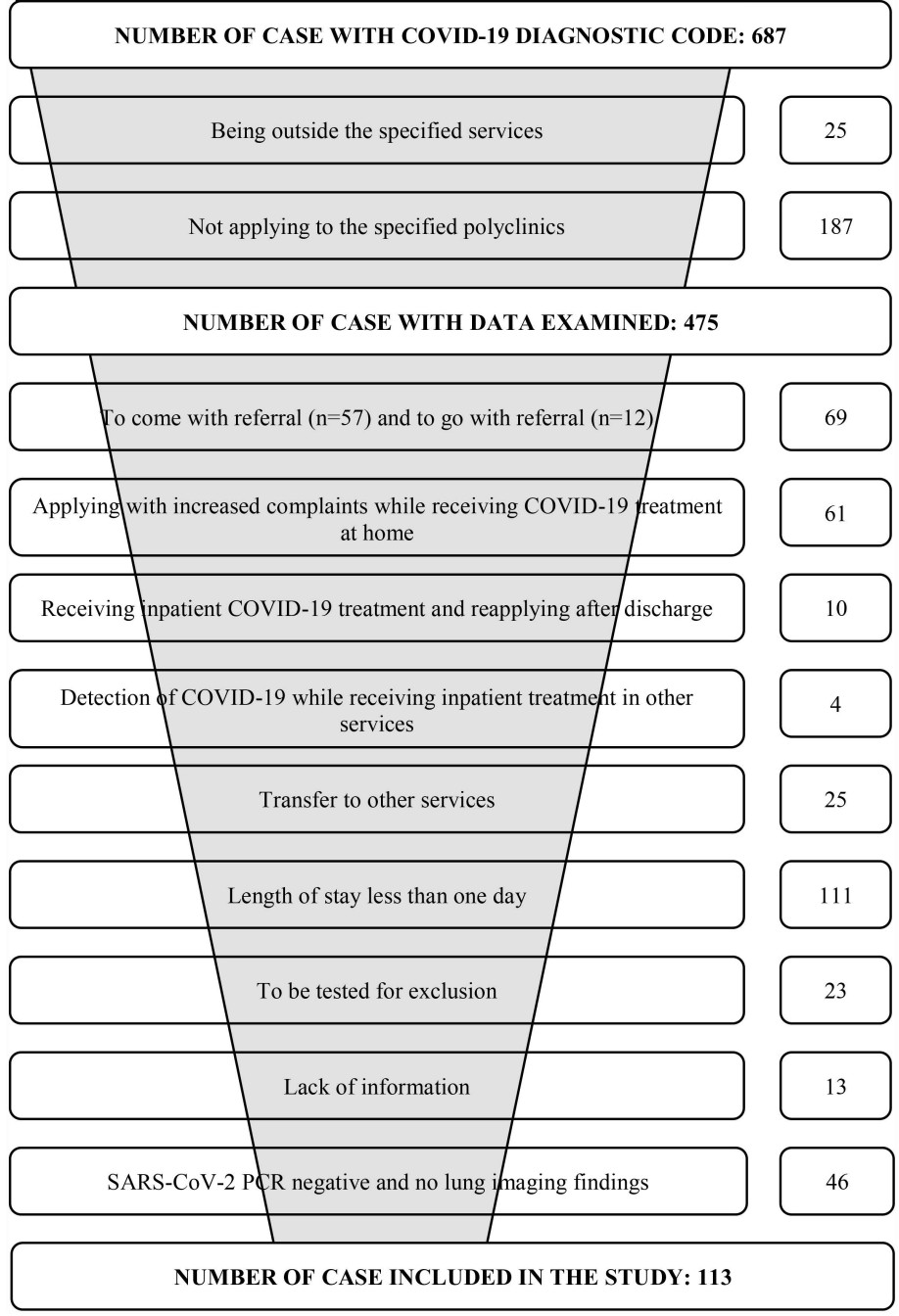

**Fig 1. Detection, selection and exclusion steps of the cases.**

## 2.4. Examined data, data collection method and data sources

Patients were identified from HIPC records. By using the anamnesis, consultation and epicrisis notes on HIMS, the sociodemographic and personal characteristics of the patients, their health status characteristics, the information about the diagnosis and treatment services they received in the hospital were examined manually for each patient. Information on patient costs was obtained from HIPC in MS Excel format. Costs are calculated over the unit cost of each transaction. Physician examinations, laboratory tests, radiological imaging methods, therapeutic procedures (general applications and interventions, medical applications, anesthesia and reanimation applications, surgical applications) and bed costs are based on the unit costs included in ANNEX-2/B of the Health Practice Communique. Unit costs of drugs, non-drug therapeutic products (blood products, immune plasma, nutritional products, fluids) and consumables are based on hospital purchasing costs. There was no change in the pricing of transaction costs during the data collection period of the study. The data of the study did not require adjustment for inflation [14,15].

## 2.5. Case definitions

Diagnosis and treatment guidelines used in the management of the disease are also updated with the changes in the clinic of the disease over time. Case definitions (3) and definitions of imaging findings (Typical, Indeterminate, Atypical and Negative CT findings) are included in the Covid-19 General Information, Epidemiology and Diagnostic Guide published by the Ministry of Health of the Republic of Turkey on December 7, 2020, in its most recent form [3,16]. The suitability of the patients included in the study with the case definitions was not evaluated separately, and it was accepted as it was in the system.

## 2.6. Variables

The dependent variables of the study are Covid-19 hospital cost, Covid-19 cost per person, daily Covid-19 cost per person and estimated Covid-19 hospital cost in Turkey. The independent variables of the study were age, gender, occupation, smoking status, presence of chronic disease, thorax CT finding, SARS-CoV-2 PCR result, presence of concomitant infection, severity of disease, length of hospital stay, type of hospitalization, oxygen inhalation and NIMV/ IMV use case.

## 2.7. Categorization

Disease severity is grouped as 'mild', 'moderate', 'severe and critical' according to the WHO [17]. Drugs are categorized according to Anatomic Therapeutic Chemical Classification (ATC) [18–21]. Mild disease refers to symptomatic patients who meet the case definition for COVID-19 without evidence of viral pneumonia or hypoxia. Moderate disease refers to adolescents or adults with SpO2 ≥ 90% on room air and clinical signs of pneumonia (fever, cough, dyspnea, rapid breathing) but no signs of severe pneumonia. Severe disease refers to adolescents or adults with clinical signs of severe pneumonia (fever, cough, dyspnea, rapid breathing) and one of the following: respiratory rate > 30 breaths/min; severe respiratory distress or SpO2 < 90% on room air. Critical disease refers to patients with acute respiratory distress syndrome (ARDS) or clinical symptoms of sepsis or septic shock [17].

Direct medical costs are costs that include the purchasable services that arise from hospital treatment. Direct medical costs, physician examination fees, laboratory tests, radiological imaging methods, therapeutic procedures (general practices and interventions, medical practices, anesthesia and reanimation practices, surgical practices), bed expenses, drugs, non-drug therapeutic products (blood products, immune plasma, nutritional products, fluids) and consumables are categorized as cost subgroups.

Duration of hospitalization did not follow normal distribution. So duration of hospitalization was categorized according to the median value, grouped as '≤8 days' and '>8 days'.

## 2.8. Calculations

**2.8.1. Age.** It was calculated by subtracting the date of birth from the hospitalization date of the cases.

**2.8.2. Duration of hospitalization.** By subtracting the date of hospitalization from the date the cases left the hospital; Length of stay in the service and ICU was obtained by subtracting the date of admission to the unit from the date of leaving the relevant unit. Presented as a day.

**2.8.3. Currency.** Costs are calculated in Turkish Lira (TL), US Dollars (USD) and Euros (EUR). The amount of each cost item has been converted from TL to USD and EUR. In the calculation of costs in USD and Euro; The exchange rates specified in the Indicative Central Bank Rates published daily by the Central Bank of the Republic of Turkey on the date of the patient's hospitalization were used [22].

**2.8.4. Costs per person.** Calculated by dividing the sum of the expenditures made by the hospital for the cases examined in a one-year period by the number of cases.

**2.8.5. Per person per day costs.** The expenditure made by the hospital for each case is divided by the number of hospitalization days of the case. The mean daily cost per person was calculated by taking the arithmetic average of these values for each patient.

**2.8.6. Covid-19 hospital cost.** It is the sum of the expenses made by the hospital for the diagnosis and treatment of all cases of Covid-19.

**2.8.7. Estimation of costs of patients receiving Covid-19 treatment in Turkey.** The total of the direct medical costs of inpatients receiving Covid-19 treatment across Turkey was estimated using the 'total number of hospitalized patients' announced by the Ministry of Health. According to the report named Covid-19 Weekly Status Report, 19.10.2020 - 25.10.2020, Turkey', this number is 172.346 as of 25.10.2020. In the study, the mean cost per person and the median of 69 cases whose hospitalization was completed before 25.10.2020 were calculated [23]. This mean and median were separately multiplied by the 'total number of hospitalized patients' included in the status report.

## 2.9. Analysis

Data were analyzed using Microsoft (MS) Excel and SPSS 23.0 package program. Descriptive statistics; number and percentage for categorical variables; are given as mean, standard deviation, median, minimum and maximum for continuous variables. In the literature, numerical values are given as median in some studies and as mean in some studies. To facilitate comparison of studies, we presented numerical values together with both median and mean values. Conformity of continuous variables to normal distribution was tested with the One Sample Kolmogorov - Smirnov Test. Mann Whitney U Test, Kruskal Wallis Analysis of Variance (advanced posthoc analysis with Bonferroni correction when necessary) and Spearman Test were performed for continuous variables that did not fit the normal distribution. Chi-square test was used to compare categorical variables. In the multivariate analysis, logistic regression analysis was used to identify independent predictors of Covid-19 costs per capita. The dependent variable for the analysis, the cost of Covid-19 per person, was categorized from its median value. The variables that were found to be important in the univariate analyzes and the variables that were stated as risk factors for Covid-19 in the literature were included in the model. Age, gender, presence of chronic disease, SARS-CoV-2 PCR result, severity of the disease, receiving oxygen inhalation therapy, and length of hospital stay (days) were

the independent variables. The model was established with the backward method. Hosmer-Lemeshow test was used for model fit, and Nagelkerke R2 and Cox - Snell R2 tests were used to evaluate explanatory power. Odds Ratio (OR) values are presented with 95% Confidence Interval (CI). Statistical significance level was accepted as p < 0.05.

## 2.10. Ethical statement

Ethical approval was obtained from the Scientific Research Ethics Committee of Karadeniz Technical University Faculty of Medicine (Code number: 2021/93). As this study is an archive based retrospective study, a permission to use the relevant data of patients was given from Directorate of Karadeniz Technical University Farabi Hospital (Code Number: 87993219-619-14316). The relevant data was fully anonymized by the Directorate of Karadeniz Technical University Farabi Hospital Information Management Unit before the authors reached them. The data was accessed for research purposes between 01.05.2021 – 01.08.2021.

## 3. Results

In this study, the direct medical costs of 113 Covid-19 cases over the age of 18 who received inpatient treatment at KTUFMFH during the one-year period since the declaration of the pandemic were examined from the hospital perspective.

Covid-19 hospital cost was calculated as 62,080.51 USD, the median cost of Covid-19 per person for cases was 263.55 (min: 28.30 – max: 18,947.60) USD, and the daily median cost of Covid-19 per person was 28.49 (min: 11.58 – max: 473.69) USD. In estimating the total costs of the patients who received inpatient Covid-19 treatment throughout Turkey, the per capita costs of 69 patients whose hospitalizations were between 13.03.2020 and 25.10.2020 were calculated and multiplied by the total number of hospitalized patients (n = 172,346). The median and mean values of COVID-19 costs per person for 69 cases were found to be 200.66 (min: 28.30 – max: 2,361.62) and 276.82 ± 329.11 USD, respectively. The total estimated hospital costs of the patients who received Covid-19 treatment in Turkey between 13.03.2020 and 25.10.2020 were calculated as 47,709,171.72 and 34,582,963.96 USD, respectively, over the median and average values (Table 1).

**Table 1. Covid-19 hospital cost, Covid-19 costs per case, per case per day Covid-19 costs and estimated Covid-19 hospital cost in Turkey between 13.03.2020 – 25.10.2020.**

| | | TL | USD | EUR |
|---|---|---|---|---|
| Covid-19 Hospital Cost (n = 113) | Total | 465,418.53 | 62,080.51 | 52,669.52 |
| Covid-19 Costs Per Case (n = 113) | Mean ± SD | 4,118.75 ± 14,133.52 | 549.39 ± 1,802.01 | 466.10 ± 1,486.34 |
| | Median | 1,834.20 | 263.55 | 229.55 |
| | Min - Max | 206.43 – 148,685.62 | 28.30 – 18,947.60 | 23.90 – 15,633.67 |
| Per Case Per Day Covid-19 Costs (n = 113) | Mean ± SD | 361.57 ± 446.00 | 49.39 ± 57.99 | 42.54 ± 48.23 |
| | Median | 208.08 | 28.49 | 25.47 |
| | Min - Max | 90.89 – 3,717.14 | 11.58 - 473.69 | 9.56 - 390.84 |
| Covid-19 Costs Per Case for 69 cases | Mean ± SD | 1,940.02 ± 2,430.26 | 276.82 ± 329.11 | 247.93 ± 278.26 |
| | Median | 1,341.03 | 200.66 | 183.47 |
| | Min - Max | 206.43 - 17,280.93 | 28.30 – 2,361.62 | 23.90 – 1,997.31 |
| Covid-19 Hospital Cost in Turkey[*] | Multiply by the mean | 334,354,318.44 | 47,709,171.72 | 42,729,253.94 |
| | Multiply by the median | 231,121,707.89 | 34,582,963.96 | 31,620,072.71 |

[*]Estimated Covid-19 Hospital Cost in Turkey Between 13.03.2020 - 25.10.2020 (n = 172,346).

The distribution of Covid-19 hospital costs of 113 cases included in the study by cost subgroups was analyzed. It was observed that 32.01% of the cost was drugs and 22.60% was laboratory tests (Table 2). It has been observed that 70.38% of the total cost of drugs is 'systemic anti-infectives', 7.12% are 'nervous system', 6.30% are 'drugs for blood and blood-forming organs'. It was determined that 86.30% of the cost of systemic anti-infective drugs was composed of 'immune serum and immunoglobulins', 12.54% of 'antibacterials' and 1.16% of 'antivirals'.

Covid-19 costs per person were examined according to the sociodemographic, health status and clinical characteristics of the cases. Median costs per capita by age groups were statistically significantly higher in the 3rd and 4th quartiles than in the 1st and 2nd quartiles ($p < 0.001$). It was observed that the costs were similar according to gender, occupational group (with and without health workers), smoking status. The median cost per person was statistically significantly higher in patients with any chronic disease than in those without ($p < 0.001$). There was a statistically significant difference in per capita Covid-19 costs between those with typical, vague, atypical, and negative thoracic CT findings ($p = 0.032$). No significant difference was found between the groups in posthoc analyses. The median costs of Covid-19 per person were statistically significantly higher in those with a positive SARS-CoV-2 PCR result ($p < 0.001$). The median cost of Covid-19 per capita was found to be statistically significantly higher in cases with accompanying infectious disease compared to cases without concomitant infectious disease ($p < 0.001$). The median costs of Covid-19 per capita differed statistically between mild, moderate, severe, and critically ill patients ($p < 0.001$). In posthoc analysis, costs were found to be significantly higher in the severe and critical case groups compared to the mild and moderate groups ($p < 0.001$ and $< 0.001$, respectively). The median costs of Covid-19 per person, in patients who received oxygen inhalation therapy, compared to those who did not; who received NIMV and/or IMV treatment, compared to those who did not, who hospitalized in the ICU and/or service, compared to those who were hospitalized only in the service; were found to be statistically significantly higher who died than those who were discharged ($p < 0.001$; $< 0.001$; $< 0.001$ and $< 0.001$ respectively). The median hospital stay of the cases was calculated as 8.0 (min:1.0 – max:40.0) days. The median costs of Covid-19 per capita were found to be statistically significantly higher in patients with a hospital stay of more than 8 days compared to those with a hospital stay of 8 days or less ($p < 0.001$) (Table 3).

**Table 2. Distribution of the Covid-19 hospital cost, per case costs of Covid-19 and per case per day costs of Covid-19 by subgroups of costs (USD, n = 113).**

| Cost subgroup | Percentage in Hospital Cost (%) | Sum of Covid-19 Costs Per Case | n | Costs Per Case (USD) | | | Cost Per Case Per Day (USD) | | |
|---|---|---|---|---|---|---|---|---|---|
| | | | | Mean ± SD | Median | Min - Max | Mean ± SD | Median | Min - Max |
| **Physician Examinations** | 1.39 | 890.67 | 112 | 7.95 ± 6.37 | 6.23 | 2.17 - 40.26 | 1.21 ± 1.67 | 0.75 | 0.17 - 12.90 |
| **Laboratory Examinations** | 22.60 | 14,461.05 | 113 | 127.97 ± 113.14 | 105.75 | 15.28 - 955.14 | 15.45 ± 10.70 | 13.07 | 1.92 - 86.78 |
| **Radiological Imaging** | 2.69 | 1,741.17 | 108 | 16.12 ± 11.53 | 10.34 | 0.87 - 72.50 | 2.95 ± 7.14 | 1.50 | 0.12 - 72.50 |
| **Therapeutic Procedures** | 11.26 | 7,076.81 | 113 | 62.63 ± 87.71 | 42.48 | 4.02 - 785.70 | 6.69 ± 5.73 | 4.52 | 0.89 - 31.46 |
| **Bed Fee** | 10.43 | 6,655.96 | 113 | 58.90 ± 68.39 | 43.35 | 4.52 - 583.14 | 5.86 ± 2.95 | 4.87 | 4.04 - 16.10 |
| **Drugs** | 32.01 | 19,292.56 | 113 | 170.73 ± 1,259.66 | 19.98 | 0.91 – 13,397.14 | 8.91 ± 32.91 | 2.26 | 0.18 - 334.93 |
| **Non-drug Therapeutic Products** | 12.22 | 7,467.10 | 79 | 94.52 ± 276.71 | 9.88 | 0.32 – 2,216.14 | 6.81 ± 12.29 | 1.53 | 0.02 - 55.40 |
| **Consumables** | 7.39 | 4,495.18 | 94 | 47.82 ± 161.00 | 3.58 | 0.12 – 1,080.65 | 4.44 ± 15.37 | 0.67 | 0.01 - 115.18 |
| **Covid-19 Hospital Cost** | **100.00** | **62,080.51** | **113** | **549.39 ± 1,802.01** | **263.55** | **28.30 – 18,947.60** | **49.39 ± 57.99** | **28.49** | **11.58 - 473.69** |

**Table 3. Covid-19 costs per case by characteristics.**

| | n | % | Costs Per Case (USD) | | | |
|---|---|---|---|---|---|---|
| | | | Mean ± SD | Median | Min - Max | p |
| **Age Group (years)** | | | | | | |
| First Quarter (22.9 - 38.5) | 28 | 24.8 | 212.43 ± 98.05 | 192.52 [a] | 51.07 - 465.32 | <0.001 |
| Second Quarter (38.6 - 56.9) | 29 | 25.6 | 269.81 ± 289.72 | 200.66 [a] | 28.30 – 1,537.06 | |
| Third quarter (57.0 - 71.1) | 28 | 24.8 | 1202.62 ± 3517.79 | 335.46 [b] | 130.92 – 18,947.60 | |
| Fourth quarter (71.2 - 96.5) | 28 | 24.8 | 522.67 ± 594.63 | 339.16 [b] | 100.55 – 2,593.17 | |
| **Presence of Chronic Disease** | | | | | | |
| Yes | 62 | 54.9 | 528.26 ± 554.90 | 325.16 | 55.47 – 2,593.17 | <0.001 |
| No | 51 | 45.1 | 575.07 ± 2,626.21 | 186.49 | 28.30 – 18,947.60 | |
| **Thorax CT Finding** | | | | | | |
| Typical | 55 | 52.9 | 838.57 ± 2,552.96 | 311.72 [a] | 51.07 – 18,947.60 | 0.032 |
| Uncertain | 25 | 24.0 | 252.18 ± 161.09 | 232.47 [a] | 76.06 - 895.53 | |
| Atypical | 10 | 9.6 | 237.46 ± 162.27 | 157.39 [a] | 74.36 - 556.91 | |
| Negative | 14 | 13.5 | 262.16 ± 141.57 | 241.93 [a] | 100.55 - 624.47 | |
| **SARS-CoV-2 PCR Resulting** | | | | | | |
| Positive | 65 | 57.5 | 777.89 ± 2,350.52 | 298.28 | 28.30 – 18,947.6 | <0.001 |
| Negative | 48 | 42.5 | 239.96 ± 211.64 | 200.29 | 51.07 – 1,537.06 | |
| **Presence of Concomitant Infection** | | | | | | |
| Yes | 12 | 10.6 | 2,490.93 ± 5,247.36 | 790.73 | 232.47 – 18,947.60 | <0.001 |
| No | 101 | 89.4 | 318.71 ± 319.37 | 232.01 | 28.30 – 2,277.46 | |
| **Severity of Disease** | | | | | | |
| Mild | 17 | 15.1 | 176.17 ± 69.16 | 167.04 [a] | 74.36 - 275.85 | <0.001 |
| Middle | 64 | 56.6 | 243.40 ± 134.81 | 218.16 [a] | 28.30 - 781.87 | |
| Severe and Critical | 32 | 28.3 | 1,359.62 ± 3,278.00 | 500.14 [b] | 162.55 – 18,947.60 | |
| **Receiving Oxygen Inhalation** * | | | | | | |
| Yes | 43 | 38.1 | 1,061.74 ± 2,860.04 | 417.58 | 100.55 – 18,947.60 | <0.001 |
| No | 70 | 61.9 | 234.65 ± 166.71 | 198.78 | 28.30 – 1,124.10 | |
| **Receiving NIMV/ IMV** | | | | | | |
| Yes | 13 | 11.5 | 2,650.34 ± 4,953.74 | 1,264.24 | 437.20 – 18,947.60 | <0.001 |
| No | 100 | 88.5 | 276.26 ± 210.23 | 229.95 | 28.30 – 1,537.06 | |
| **Place of Hospitalization** | | | | | | |
| Service Only | 97 | 85.8 | 257.09 ± 162.06 | 224.29 | 28.30 – 1,124.1 | <0.001 |
| ICU/ ICU and Service | 16 | 14.2 | 2,321.41 ± 4,491.79 | 1,079.88 | 417.58 – 18,947.60 | |
| **Long of Hospitalization (Days)** | | | | | | |
| ≤8 | 57 | 50.4 | 211.23 ± 155.95 | 165.59 | 28.30 - 895.53 | <0.001 |
| >8 | 56 | 49.6 | 893.58 ± 2,519.66 | 338.96 | 150.57 – 18,947.60 | |
| **End of Hospitalization** | | | | | | |
| Discharge | 98 | 86.7 | 284.22 ± 236.36 | 229.95 | 28.30 – 1,537.06 | <0.001 |
| Death | 15 | 13.3 | 2,281.78 ± 4,677.95 | 895.53 | 162.55 – 18,947.60 | |

*Receiving Oxygen Inhalation: with any of mask, reservoir mask, HFNO, NIMV or IMV.

In post hoc analyses there is a significant difference between the different letters in the columns as indicated.

Age, gender, presence of chronic disease, PCR result, severity of the disease, receiving oxygen inhalation therapy, and length of stay in the hospital were included in the logistic regression model to determine the factors affecting the per capita cost of Covid-19. According to the model; it was found that the cost was 2.60 times higher with a one-day increase in

hospitalization, 15.13 times in the presence of chronic disease, and 15.24 times when oxygen inhalation therapy was taken (p < 0.001, 0.016, 0.013, respectively). When mild severe disease is taken as reference, it was seen that the cost increased 77.48 times in severe and critical cases (p = 0.006) (Table 4).

## 4. Discussion

Understanding the costs is important in managing Covid-19. As a result of scientific studies, the diagnosis and treatment algorithms of the disease are updated. Accordingly, the true economic dimension of the disease continues to change continuously.

In our study, it was determined that the Covid-19 hospital cost of 113 patients who were hospitalized at KTUFMFH and diagnosed with Covid-19 in the first year of the pandemic was 62,080.51 USD. The estimated total hospital costs of patients receiving inpatient treatment due to Covid-19 between 13.03.2020 and 25.10.2020 were calculated as 34,582,963.96 USD. In our study, the median and mean per capita cost of Covid-19 were calculated as 263.55 and 549.39 ± 1,802.01 (min: 28.30 - max: 18,947.60) USD [229.55 and 466.10 ± 1,486.34 (min: 23.90 - max: 15,633.67) EUR], respectively. In our study, the median and mean hospital stays of the patients were found to be 8.0 and 9.5 ± 6.1 (min: 1.0 - max: 40.0) days, respectively. 28.3% of the patients had severe or critical illness, while 14.2% were treated in intensive care.

A study by Miethke-Morais in Brazil found that the total hospital costs of 3,254 hospitalizations in March - June 2020 were $41,122,173, with an average direct medical cost per person of $12,637 [7]. In this study, the rate of comorbid diseases and intensive care admission rates are quite high compared to our study.

A study by Li et al. examining the costs of 70 inpatient hospitalizations in China from January to March 2020 estimated total hospital costs at $494,202,120 based on the total number of Covid-19 cases in China (81,416) as of March 21, 2020. The study found an average cost of $6,827 per person; the average length of hospital stay was 16 days [6]. A study by Liang et al. found that the total cost of 220 suspected or confirmed inpatient hospitalizations in China from January to April 2020 was $814,326. The average cost per person in their study was

**Table 4. Regression analysis of factors associated with Covid-19 cost per case.**

|  | B | Wald | p | OR | %95 CI | |
|---|---|---|---|---|---|---|
| **Presence of Chronic Disease** |  |  |  |  |  |  |
| No |  |  |  | 1 |  |  |
| Yes | 2.717 | 5.756 | 0.016 | 15.130 | 1.644 | 139.216 |
| **Severity of Disease** |  |  |  |  |  |  |
| Mild |  |  |  | 1 |  |  |
| Middle | 0.605 | 0.440 | 0.507 | 1.831 | 0.306 | 10.946 |
| Severe and Critical | 4.350 | 7.692 | 0.006 | 77.476 | 3.582 | 1,675.670 |
| **Receiving Oxygen Inhalation Therapy** |  |  |  |  |  |  |
| No |  |  |  | 1 |  |  |
| Yes | 2.724 | 6.195 | 0.013 | 15.238 | 1.784 | 130.157 |
| **Long of Hospitalization (Days)** | 0.955 | 13.997 | <0.001 | 2.600 | 1.576 | 4.289 |

**Variables included in the model:** Age, sex, presence of chronic disease, SARS-CoV-2 PCR result, severity of disease, receiving oxygen inhalation therapy, long of hospital stay (days).

Hosmer – Lemeshow: 0.854.

Nagelkerke $R^2$: 0.797 ve Cox – Snell $R^2$: 0.598.

**Omnibus:** < 0.001.

$2,158; the average length of hospital stay was 18 days [24]. In the study conducted by Jin et al., the average treatment cost of a Covid-19 patient with a positive PCR test was estimated to be 3,193 US dollars, according to literature, expert opinion, and information in clinical guidelines in China. This study used the bottom-up costing method [10]. The reason why the costs were higher in China may be that the hospital stay in these studies was longer and all Covid-19 patients were treated as inpatients according to Chinese guidelines.

In Ghaffari Darab's research examining the costs of 477 Covid-19 patients hospitalized in a university hospital in Iran between March and July 2020, it was determined that the total direct medical costs of the patients were $1,791,172, the average cost per person was $3,755, and the average hospital stay was 7 days [25]. In Ebrahimipour's study, the average cost per person for 2,980 Covid-19 cases hospitalized in Iran was $439; the average hospital stay was 7 days [26].

Bozdemir et al., in their study examining the direct medical costs of 582 Covid-19 patients who received inpatient treatment in Turkey from March to December 2020 from the perspective of the Social Security Institution, found that the total hospital cost was US$1,052,595, and the average cost per person was US$1,808. The average hospital stay was 5.7 days [27]. 23% of the patients were admitted to intensive care. This rate was 14.2% in our study. Bozdemir's study, unlike our study, is from an insurance perspective. In our country, social insurance was making package payments for Covid-19, not individual costs. In our study, the cost was obtained by adding up each cost item. In addition, the intensive care admission rates in the study in question are higher than in our study. In addition, the sample is much larger than ours. This also increases the costs. Gedik's average per capita costs of 459 patients in the ward and intensive care unit in March - May 2020 were 882 ± 667 and 2,924 ± 2,347 US dollars, respectively. The average length of stay in the ward and intensive care unit in the study was 9.0 and 14.7 days, respectively, and the intensive care admission rate was 24% [28]. Their longer hospital stays and higher rates of intensive care admission may explain why their studies are more expensive than ours.

In Carrera-Hueso et al.'s study of 254 SARS-CoV-2 PCR positive inpatient facilities, the average cost per person for those not in ICU was €50,132 and for those in ICU was €280,956. These hospital stay and ICU stays were 44.1 ± 4.8 days and 37.8 ± 4.3 days, respectively [29]. The fact that their study included staff salaries and the length of hospital stay was much longer than in our study explains why the costs were higher than in our study.

Ohsfelt's study in the USA between April and October 2020 found that the median hospital cost of 173,942 inpatients was 12,046 USD. The median hospital stay in this study was 5 days. The intensive care rate in this study is significantly higher than in our study [30]. In a study conducted by Fusco in the USA between April and December 2020, it was found that the median hospital cost of 198,806 inpatients was 1,267 USD. The median hospital stay in this study was 6 days. The rate of intensive care patients in this study is quite high compared to our study. In addition, approximately half of the population consists of elderly patients [31]. The increase in comorbidities with age and the increase in treatment costs may explain the higher costs compared to our study. In addition, the higher costs in these studies may be due to the increase and standardization of diagnostic and treatment opportunities over time and the decrease in costs compared to the emergence of Covid-19.

A study by Lee et al estimated that the average cost per person for 145 pediatric patients hospitalized in Korea between February and March 2020 was US$2,192; the average hospital stay was 10 days; and the average cost per person for mild to moderate adult cases was approximately US$3,916 [32].

In a study by Khan et al, the average direct medical cost per person for 1,422 inpatients in the Kingdom of Saudi Arabia was US$12,916, and the median length of stay was 8 days [33].

The study included salaries for medical staff, which could explain the higher costs. Reddy's study found that for 176 patients admitted to the ICU in India, the average cost per person was $3,192; the median hospital stay was 13 days [34]. Most patients had comorbidities and many patients died. This may be associated with increased costs.

Private health insurance is widespread in the US in terms of organization and financing. Expenses are very high. In China, financing is provided by taxes, premiums and direct payments. Privatization is increasing in China's health system. Turkey has a social insurance system. It is mostly public. Premiums and taxes are collected. Expenses are low.

The number of cases increased at different times in different parts of the world during the pandemic. Characteristics such as severity of illness, average length of hospital stay, and length of stay in the intensive care unit also differentiated costs across study populations. The difference between these studies may be due to increased and standardized diagnostic and therapeutic opportunities and decreased costs over time compared to the emergence of Covid-19.

Costs consist of various cost subgroups. Costs are determined from the perspective of the study. Access to cost data is important in determining these costs. In our study, drugs accounted for 32.01% of the Covid-19 hospital costs and laboratory tests accounted for 22.60%. Study of Bozdemir et al. 79% of the cost consisted of the intervention cost [27]. In studies in Iran, the majority of hospital costs consisted of bed fees, drugs and medical consumables [25, 26]. In the study in China, approximately half of the hospital costs were drugs; In the study in Brazil, most of it consisted of personnel payments, followed by medicines and consumables [6, 7]. In a study in Spain, almost all of the hospital costs of PCR-positive patients consisted of health care and hospitalization fees, including physician and nurse salaries [29]. In studies where the salaries of healthcare workers are also taken into consideration, salaries constitute the largest part of the cost, followed by costs related to drugs and beds. In studies where personnel salaries are not included, drug costs have an important place. There are no agent-specific drugs in the treatment of Covid-19 yet. Therefore, the combination of more than one drug with limited evidence or symptomatic treatment may be necessary. Moreover in cases with chronic diseases, drugs for the treatment of existing diseases are also used. With the combined use of drugs with different mechanisms of action, drug side effects may occur and drug changes can be made. These situations increase the costs of drugs.

In our study, median costs per person were found to be significantly higher in older age groups. In studies in the USA and China, the average cost per person of patients in the older age group was found to be significantly higher [6,24,30]. In the study of Miethke-Morais et al., the costs were found to be 1.51 times higher in the age group above the age of 69, when the costs were taken as reference under the age of 18 [7]. In a study conducted on patients hospitalized in the ICU, the costs were similar by age groups [34]. Aging is associated with decreased immune response. This makes people prone to both getting an infection and a more severe course of the disease, and may prolong the treatment process. In addition, costs may increase due to the fact that chronic diseases are more common as age progresses.

In our study, it was determined that the presence of chronic disease increased the costs 15.13 times. In studies in Brazil and China, hospital costs were found to be significantly higher in patients with chronic diseases [6, 7]. In the study of Miethke-Morais et al., the costs were found to be 1.24 times higher in patients with 3 or more comorbidities than those without [7]. In a study conducted in India, the costs were significantly higher in patients with diabetes mellitus than in patients without [34]. In the study of Ohsfeldt et al., being diagnosed with asthma, diabetes or obesity was associated with significantly higher costs (compared to the absence of related diagnoses) [30]. Chronic diseases can increase hospital costs by prolonging the length of stay by requiring physician examinations, drug use, examination and use of medical devices.

Studies have reported that SARS-CoV-2 viral load among hospitalized Covid-19 patients is independently associated with the risk of intubation and in-hospital death [35]. In our study, the median costs of Covid-19 per person were found to be significantly higher in those with positive SARS-CoV-2 PCR results compared to those with negative results. Miethke-Morais et al. In the study, the costs were found to be 1.61 times higher in patients with positive SARS-CoV-2 PCR results [7]. In cases with positive SARS-CoV-2 PCR test results, the disease may progress more severely due to high viral load, which may increase costs.

Co-infections in patients with Covid-19 are associated with adverse health outcomes [36]. In a meta-analysis, it was reported that 6.9% of patients hospitalized due to Covid-19 were found to have a bacterial infection agent [37]. The effect of this situation on the cost of the disease was not found in the literature reviewed. In our study, the median cost of Covid-19 per person was found to be significantly higher in cases with accompanying infectious disease compared to those without concomitant infectious disease. Concomitant infectious disease may increase the patient's examination, diagnostic tests, the use of drugs for the agent and/or symptoms, the use of different medical interventions and the use of medical devices by aggravating the patient's clinic. It can also increase costs by requiring a longer hospital stay.

In the regression analysis of our study, it was calculated that having a 'severe and critical' disease increased the costs per capita 77.48 times when the 'mild' disease was taken as reference. In studies in China, costs were found to be significantly higher in patients with more severe disease than in those with milder disease [6,24]. In the study of Liang et al., the development of severe and critical illness was found to be associated with significantly higher costs [24]. In cases where the clinic is severe, the increase in the use of examinations, medical devices, drugs and other therapeutic products, the need for an intensive care unit and the prolonged hospitalization may cause increased costs.

Admission of Covid-19 patients to the intensive care unit imposes a significant direct medical cost [34]. In our study, the median cost of Covid-19 per person in ICU and/or service hospitalizations was statistically significantly higher than in patients hospitalized only in the service. Studies have found that staying in the ICU is associated with significantly higher costs [24,26]. ICU admission leads to higher costs associated with more severe disease.

In our study, it was calculated that a one-day increase in the length of hospital stay increased the cost per person by 2.60 times. In a study conducted in India, the costs were found to be significantly higher in patients with long hospital stays compared to those with shorter hospital stays [34]. It is inevitable that long hospital stays increase the costs by causing more procedures such as bed cost and examination. In addition, long hospital stays are associated with the presence of chronic disease and the severity of Covid-19, and may increase costs due to the need for more investigations and therapeutic procedures.

In our study, the costs were found to be statistically significantly higher in patients who lost their lives compared to those who were discharged. In the study conducted in India, the cost of patients whose hospitalization ended with discharge was found to be significantly higher than those that resulted in death [34]. In the study of Ohsfeldt et al., it was found that hospitalization resulting in discharge home was associated with significantly lower costs [30]. In the study of Miethke-Morais et al., hospitalization resulted in death was associated with significantly lower costs (O.R.: 0.764) [7]. This difference between studies may be due to the difference in intensive care facilities available to keep the patient alive.

## 4.1. Strengths and limitations

Our study has some strengths. As far as we could detect, our study is the study with the largest time interval. Because it covers hospitalizations during the first year of the pandemic. The study was conducted with realized costs. In our study, all of the cost items were analyzed one

by one and categorized using the bottom-up method. In this way, it was possible to examine in detail the resources used in the management of the disease. In the study, data on both diagnosis and treatment expenses were obtained. In the conversion of cost data to USD and EUR, daily exchange rates are used instead of mid-year, in order to better reflect the reality. Our study is one of the few studies where regression analysis was performed to identify the main drivers of costs.

Our study has some limitations. These should be considered when interpreting the results of the study. Since the study was conducted retrospectively, it was not possible to use a structured standard data collection form. Some costs may have been covered out of pocket during the hospitalization of the cases. It is not known whether the cases received inpatient treatment in another health institution after discharge. In our study, the cost of drugs was calculated on the basis of their purchasing costs. Acquisition costs can vary between institutions and lead to potential bias in estimating the national financial burden.

Since the dependent variable, the per capita cost of COVID-19, did not comply with normal distribution. Therefore, in the multivariate logistic regression analysis, the per capita cost of COVID-19 was categorized from the median value. As in other studies, the confidence intervals in our study are wide. This may be due to the reflection of changes in disease severity on costs.

During the study period, diagnosis and treatment algorithms changed frequently, which differentiated the costs. The true economic dimension of the disease is constantly changing. In this regard, it should be noted that these data in the first year of the pandemic are evaluated with today's information.

Our study provides important information about the direct medical costs of inpatient Covid-19 cases. This information provides basic data for future economic evaluation analyses.

## 5. Conclusion

The direct medical costs of 113 Covid-19 cases over the age of 18 who received inpatient treatment at KTUFMFH in the first year of the pandemic were examined from the hospital perspective. The median cost of Covid-19 per person was 263.55 (min: 28.30 – max: 18,947.60) USD and the daily median cost of Covid-19 per person was 28.49 (min: 11.58 – max: 473.69) USD.

The median costs of Covid-19 per person are significantly higher in patients who are in older age groups, have positive SARS-CoV-2 PCR results, have a more severe clinical course, are accompanied by another infectious disease, receive oxygen inhalation therapy, have been treated in the ICU for all or part of their hospitalization, have a longer hospital stay, with chronic diseases and hospitalizations resulting in death.

In our study, it was observed that Covid-19 constitutes a significant economic burden for our hospital. In reducing costs such as drugs and examinations, which constitute a significant part of the costs, it will be beneficial to choose the lower cost medicinal products among the medicinal products with the same effect and to determine the criteria for the examinations.

Early interventions are of great importance in the management of the disease and in reducing the costs. It is possible to have a milder course of the disease with early diagnosis and effective treatment of the cases. Thus, the need and duration of patients for medical procedures, examinations, therapeutic products and hospitalizations/ ICUs are reduced. This helps to lower the costs of the disease.

In mass situations such as epidemics, priorities in the management of health services, the demand for health services of the public and the use of medical products are changing. For this reason, hospitals should have action plans ready, with a proactive approach, to be put into practice without delay in case of an epidemic. Pandemic action plans should be created for

disease groups according to the transmission routes. Personnel management, use of medical devices, and less costly procurement of purchasable resources should be mentioned here. Thus, it may be possible to intervene in the epidemic earlier, and to affect less people from the disease, as well as to lower costs.

Prevention of diseases is becoming more and more important both with the protection of public health and with its effects on the economy. General public health measures should be strictly enforced in all circumstances. In addition, one of the most effective preventive methods for susceptible individuals is effective and rapid vaccination. Vaccines reduce catching the disease, complications related to the disease, hospitalization and ICU admissions, severe course of the disease and deaths. With these effects, Covid-19 vaccines will make a significant contribution to reducing the costs of the disease.

Non-communicable diseases have an important place among the risk factors of Covid-19. In our study, it was found that the costs increased in patients with chronic diseases. Taking these diseases under control in the long term will both increase the health level of the people and play an important role in preventing the increase in costs in future infectious disease outbreaks.

## Supporting information

**S1 File. Data – 2025.**
(XLSX)

## Author contributions

**Conceptualization:** Medine Gözde Üstündağ, Nazım Ercüment Beyhun, Murat Topbaş, Sevil Turhan.

**Data curation:** Medine Gözde Üstündağ.

**Formal analysis:** Medine Gözde Üstündağ.

**Methodology:** Medine Gözde Üstündağ, Nazım Ercüment Beyhun.

**Supervision:** Nazım Ercüment Beyhun.

**Visualization:** Medine Gözde Üstündağ.

**Writing – original draft:** Medine Gözde Üstündağ, Nazım Ercüment Beyhun, Murat Topbaş, Sevil Turhan.

**Writing – review & editing:** Medine Gözde Üstündağ, Nazım Ercüment Beyhun, Murat Topbaş, Sevil Turhan.

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
