## [Decision Letter · Decision Letter 0]

16 Jan 2025

PONE-D-24-49804Direct Medical Cost of Adult COVID-19 Inpatients and Its Determinants at a University HospitalPLOS ONE

Dear Dr. Beyhun,

Thank you for submitting your manuscript to PLOS ONE. After careful consideration, we feel that it has merit but does not fully meet PLOS ONE’s publication criteria as it currently stands. Therefore, we invite you to submit a revised version of the manuscript that addresses the points raised during the review process.

We look forward to receiving your revised manuscript.

Kind regards,

Meryem Merve Ören Çelik

Academic Editor

PLOS ONE

Journal Requirements:

“I have read the journal's policy and the authors of this manuscript have the following competing interests: [No competing interests]”

3. We note that your Data Availability Statement is currently as follows: “All relevant data are within the manuscript and its Supporting Information files.”

Please confirm at this time whether or not your submission contains all raw data required to replicate the results of your study. Authors must share the “minimal data set” for their submission. PLOS defines the minimal data set to consist of the data required to replicate all study findings reported in the article, as well as related metadata and methods (https://journals.plos.org/plosone/s/data-availability#loc-minimal-data-set-definition ).

If your submission does not contain these data, please either upload them as Supporting Information files or deposit them to a stable, public repository and provide us with the relevant URLs, DOIs, or accession numbers. For a list of recommended repositories, please see https://journals.plos.org/plosone/s/recommended-repositories .

4. Please include captions for your Supporting Information files at the end of your manuscript, and update any in-text citations to match accordingly. Please see our Supporting Information guidelines for more information: http://journals.plos.org/plosone/s/supporting-information .  

**Additional Comments:**

The manuscript provides a comprehensive analysis of COVID-19-related hospital costs, comparing findings from Turkey with international studies. It offers valuable insights into the financial implications of managing COVID-19, emphasizing differences in healthcare systems and cost structures. I thank the authors for their diligent work and contribution to this important topic. Below are some suggestions and critiques aimed at further strengthening the manuscript.

**1. Methodological Clarity and Comparisons**

• Criticism: Lack of explanation for why the study period or certain variables were selected (e.g., why "8 days" was used as a threshold for long hospital stays).

• Suggestion: Provide justification for these methodological choices in the "Materials and Methods" section, linking them to prior research or clinical standards.

• Criticism: While the study provides detailed results of cost analyses from various countries, it does not sufficiently compare or analyze these findings in a meaningful way. Differences in healthcare systems, cost calculation methods, data collection processes, and the scope of costs included are not adequately explained, which makes it challenging for readers to contextualize and interpret the findings.

• Suggestion: Expand the methodological section to include detailed descriptions of the cost calculation criteria and approaches used in the studies referenced. Explain any significant differences in methodologies and their implications for the results. This will enhance the comparability and overall value of the discussion.

**2. Contextual Differences**

• Criticism: The discussion mentions that "policies followed in the management of the disease, diagnosis and treatment algorithms, and resources in healthcare delivery may vary between countries," but this statement remains too general and does not elaborate on how these variations affect the reported costs.

• Suggestion: Incorporate concrete examples to illustrate how differences in healthcare policies, resource allocation, and treatment protocols influence costs. For instance, discuss how lower personnel costs in one country or higher intensive care unit availability in another might drive differences in healthcare expenditures.

**3. Statistical Details and Consistency in Data Presentation**

• Criticism: Some results are presented without adequate statistical support, such as p-values, confidence intervals, or post hoc analysis (especially table 3) results. This limits the rigor of the conclusions drawn.

• Suggestion: Include statistical significance metrics (e.g., p-values, confidence intervals) for all reported findings. Conduct and report post hoc analyses to clarify which groups or variables are driving observed differences. Where appropriate, include visual aids like graphs or tables to enhance clarity.

• Criticism: Inconsistencies in data presentation, such as reporting median costs in some cases and mean/ average, or range values for costs in others, make cross-study comparisons less straightforward. Additionally, numerical formatting issues (e.g., inconsistent use of commas and periods) reduce readability.

• Suggestion: Standardize the use of cost metrics (e.g., median (min-max) or mean±std dev.) across the paper. If different metrics are used, provide a rationale for these choices. Ensure that numerical formatting follows international standards (e.g., commas for thousands, periods for decimals).

• Criticism: Inconsistent use of formats for numerical values (e.g., "1,2046 USD" should likely be "1,204.6 USD"). This issue appears multiple times in cost data tables and narrative text.

• Suggestion: Standardize numerical formatting throughout the manuscript, adhering to international conventions (e.g., commas for thousands, periods for decimals).

•

**4. In-depth Discussion of Sources**

• Criticism: Discussion compares costs in Turkey with those in countries like Brazil, China, and the U.S. but does not explore how differences in healthcare systems might affect costs.

• Suggestion: Expand this discussion to include a brief overview of the healthcare reimbursement models or resource allocation strategies in these countries and how they differ from Turkey.

• Criticism: The study references numerous international works but does not critically evaluate their methodologies, sample sizes, or the scope of their analyses. This lack of critical appraisal undermines the credibility and depth of the discussion.

• Suggestion: Discuss the strengths and weaknesses of the referenced studies. Highlight sample size limitations, potential biases, and how the scope of each study might influence its findings. This evaluation will clarify the relevance of these studies to your own research.

**5. Positioning of Study Findings**

• Criticism: The paper does not clearly articulate how its findings contribute to the existing literature or what unique insights it offers. As a result, its academic impact may be diminished.

• Suggestion: Clearly state the unique contributions of the study. Highlight how the findings from Turkey compare to global trends and what these comparisons reveal about healthcare costs in different contexts. Emphasize the implications of these results for policy and practice.

**6. Structure and Flow**

• Criticism: The manuscript contains repetitive statements and lacks smooth transitions between paragraphs. For example, phrases like "Costs vary between studies" are reiterated without adding new information.

• Suggestion: Revise the text to eliminate redundancy and improve coherence. Use transitional phrases to connect paragraphs and create a logical flow of ideas throughout the discussion.

**Additional Suggestions for Improvement**

1. Terminological Clarity:

o Define key terms like "direct medical costs," "severity," and "long hospital stay" explicitly in the introduction or methods section. Ensure these terms are used consistently throughout the manuscript.

2. Comprehensive Conclusion:

o Synthesize the main findings of the study with the insights from the global comparisons. Highlight practical implications for policymakers and healthcare providers.

Reviewers' comments:

Reviewer's Responses to Questions

**Comments to the Author**

1. Is the manuscript technically sound, and do the data support the conclusions?

Reviewer #1: Yes

Reviewer #2: Yes

2. Has the statistical analysis been performed appropriately and rigorously? 

Reviewer #1: Yes

Reviewer #2: Yes

3. Have the authors made all data underlying the findings in their manuscript fully available?

Reviewer #1: Yes

Reviewer #2: Yes

4. Is the manuscript presented in an intelligible fashion and written in standard English?

Reviewer #1: Yes

Reviewer #2: No

5. Review Comments to the Author

Reviewer #1: Dear Authors,

My suggestions regarding the study titled "Direct Medical Cost of Adult COVID-19 Inpatients and Its Determinants at a University Hospital," which was sent to me for evaluation, are presented below.

Abstract

The abstract includes the aim, methodology, study period, type of study, key findings, and appropriate conclusions and recommendations. The keywords are suitable. No additional suggestions are provided.

Introduction

The introduction follows a general-to-specific flow and clearly states the study's objective. The first paragraph provides information on the general characteristics of COVID-19, its clinical course, and treatment methods, while the second paragraph focuses more specifically on the impact of the disease on healthcare systems and costs. Finally, the study's objective is clearly stated.

Materials and Methods

The materials and methods section explains in detail and in accordance with scientific principles how the study was conducted. The research methodology, scope, sample, data collection, and analysis processes are clearly outlined. In addition, topics such as ethical approval and data security are addressed, demonstrating compliance with scientific and ethical standards.

The statistical analyses used (logistic regression, Mann-Whitney U test, Kruskal-Wallis test, etc.) are consistent with the methods outlined in the study plan. The effects of factors on costs are presented meaningfully.

Results

The results section is scientifically written. Detailed analyses have been conducted in line with the methods used in the study, supported by statistical tests, and presented meaningfully. The study examines the determinants and distribution of costs in detail and explains the findings at both general and subgroup levels. The findings align with the planned methods and effectively answer the research questions and hypotheses. The analyses are detailed, systematic, and conducted in compliance with scientific standards. The findings strongly support the study's objective.

Discussion

The discussion section appears consistent with the study findings. The results obtained are compared with similar studies in the literature regarding cost determinants and distribution across subgroups, and meaningful connections are established.

Suggestions have also been presented in the "3.1. Strengths and Limitations" section. It is not appropriate for them to be under this heading.

The tables are sufficient and presented in a clear format.

The references are sufficient and appropriate.

Reviewer #2: This is a valuable article that will make significant contributions to the literature. I thank the authors for their effort. Despite having a small sample size, the study provides important insights.

Comments and Suggestions:

• There are formatting errors such as writing "COVID-19" instead of "Covid-19."

• The odds ratios (OR) presented in the abstract should be reported along with their confidence intervals.

• In the introduction, it is recommended to include brief information on the medical costs of Covid-19. The difference between outpatient and inpatient costs could be mentioned. Additionally, referencing the economic burden of Covid-19 on healthcare systems in some countries would help emphasize the significance of the topic.

• Please review the following sentence for clarity, as it could imply that hospitalized patients were excluded:

“Patients who were not registered to the services specified in KTUFMFH, whose hospitalization started after 12.03.2021, who were diagnosed with COVID-19 outside of KTUFMFH or who received inpatient treatment were not included in the study.”

• In Figure 1, it appears that 187 patients were excluded because they sought treatment in other clinics. The inclusion criteria state that patients admitted to three specific clinics were included. Why was this the case? For example, why was the pulmonology department not included? Could this introduce bias? Please address this issue in the limitations or discussion section.

• Add a subheading for Inclusion Criteria (before listing items directly).

• References 8 and 9 are not accessible and need to be updated. Additionally, a citation should be provided for "ANNEX-2/B of the Health Practice Communique."

• Why did you choose to use the median value as the cut-off for multivariable analysis? Cost could have been considered as a continuous variable. Please explain this in the limitations section.

• Confidence intervals are very wide. This should also be noted in the limitations section.

• The last paragraphs under the Strengths and Limitations heading include conclusions and recommendations. These should be moved to the Conclusion section.

Best regards

6. PLOS authors have the option to publish the peer review history of their article (what does this mean? ). If published, this will include your full peer review and any attached files.

**Do you want your identity to be public for this peer review?** For information about this choice, including consent withdrawal, please see our Privacy Policy .

Reviewer #1: No

Reviewer #2: **Yes: ** Şeyma Karaketir

---

## [Author Response · Author response to Decision Letter 0]

3 Feb 2025

Response to reviewers

First of all, we thank reviewers’ attention and suggestions about our manuscript. All of them has been carefully considered. The revisions we have made are explained below and are visable on revised manuscript with track changes.

1. References

a. References in the reference list have been updated to be accessible.

2. Methodological Clarity and Comparisons

a. Explanation of why the study period or specific variables were chosen has been added.

b. The methods section has been expanded to include detailed descriptions of costing criteria and approaches used in cited studies.

c. Significant differences in methodologies and their impact on results has been explained.

3. Contextual Differences

a. Concrete examples are included to illustrate how differences in health policies, resource allocation, and treatment protocols affect costs.

4. Statistical Details and Consistency in Data Presentation

a. Posthoc analyses were performed to understand which groups accounted for the observed differences (table 3).

b. Throughout the article, justification for using different cost metrics was provided.

c. The conformity of numerical formatting to international standards was reviewed.

5. In-depth Discussion of Sources

a. The discussion has been expanded on resource allocation strategies in these countries and how they differ from Turkey.

b. The strengths and weaknesses of the cited studies, sample sizes, and how the scope of each study might affect their findings were examined.

6. Positioning of Study Findings

a. The unique contributions of the study has been clearly stated.

b. Emphasis how the findings from Turkey compare to global trends and what these comparisons reveal about health care costs in different contexts.

7. Structure and Flow

a. Text has been revised to eliminate repetitions and increase consistency

8. Additional Suggestions for Improvement

Terminological Clarity:

a. Key terms has been clearly defined in the introduction or methods section.

Comprehensive Conclusion:

a. Conclusion has been revised to synthesize the key findings of the study with insights from global comparisons

Point to point revisions suggested by reviewer 2

a. Instead of "COVID-19" "Covid-19" was written.

b. The odds ratios (OR) presented in the summary are reported with confidence intervals.

c. Brief information has been added about the medical costs of Covid-19. References about the economic burden of Covid-19 on healthcare systems in some countries has been added.

d. A subheading for Inclusion Criteria has been added.

e. Which services were selected and why has been explained.

f. References in the reference list have been updated to be accessible.

g. The wide confidence intervals were discussed in the limitations section.

h. Explanation of why we chose to use the median value as the cut-off point for multivariate analysis has been added. (Since the dependent variable, the per capita cost of COVID-19, did not comply with normal distribution. Therefore, in the multivariate logistic regression analysis, the per capita cost of COVID-19 was categorized from the median value)

i. The last paragraphs under the heading Strengths and Limitations have been moved to the conclusion.

---

## [Editor Report · Decision Letter 1]

7 Feb 2025

Direct Medical Cost of Adult COVID-19 Inpatients and Its Determinants at a University Hospital

PONE-D-24-49804R1

Dear Dr. Beyhun,

We’re pleased to inform you that your manuscript has been judged scientifically suitable for publication and will be formally accepted for publication once it meets all outstanding technical requirements.

Kind regards,

Meryem Merve Ören Çelik

Academic Editor

PLOS ONE

---

## [Editor Report · Acceptance letter]

PONE-D-24-49804R1

PLOS ONE

Dear Dr. Beyhun,

I'm pleased to inform you that your manuscript has been deemed suitable for publication in PLOS ONE. Congratulations! Your manuscript is now being handed over to our production team.

Kind regards,

on behalf of

Dr. Meryem Merve Ören Çelik

Academic Editor

PLOS ONE